# Open-Loop Exhaust-Gas Cleaning System: Analysis of Effects Produced on Barcelona Port Water pH

Nelson Gustavo Díaz Delgado and Francesc Xavier Martínez De Osés *

Department of Engineering and Nautical Sciences, UPC-Barcelona Tech, 08034 Barcelona, Spain;
nelsnew@hotmail.com
* Correspondence: francesc.xavier.martinez@upc.edu

**Abstract:** The implementation of a 0.5% mass/mass sulphur cap in fuels used by ships has become a reality. Furthermore, regulation 14 of the MARPOL Convention–Annex VI (amended) establishes that the limit on fuel used by ships operating in a Sulphur Emission Control Area (SECA) cannot exceed 0.1% of mass/mass. To deal with these requirements while considering nature of the shipping business, which is the continuous carriage of cargo around the world, shipowners or shipping companies have few options for following this regulation. Apart from the use of a low-sulphur-content fuel is the use of an exhaust-gas cleaning system, also known as "scrubbers", as an alternative. The use of these systems, specifically the open-loop system, entails the discharge to the sea of residual water used during the cleaning process of exhaust gases from ship engines. The objective of this paper is to study the effect produced by discharging this residual water on the acidity level (pH) of Barcelona port water. This objective was achieved through the periodical sampling and analysis of Barcelona port water in collaboration with the IDAEA (CSIC) laboratory. We analysed the evolution of the pH results obtained so far and obtained an initial picture of the pH Barcelona port water situation in real time regarding the effect of wash-water discharges from open-loop scrubbers. Furthermore, this paper describes the implementation of a system which is going to improve the operation of open-loop exhaust-gas cleaning systems.

**Keywords:** open loop; water pollution; acidification; pH

## 1. Introduction

The MARPOL Convention [1] has implemented a new amendment which came into force on 1 January 2020, detailed in regulation 14 [2]—sulphur oxides and particulate matter, of Annex VI [3]—Prevention of Air Pollution from Ships. This amendment sets a ban on the carriage of non-compliant fuel oil for combustion purposes, for propulsion or operation onboard a ship. Non-compliant fuel means, after the entering into force of the above-mentioned amendment, a fuel with a sulphur content limit of 0.5% mass by mass. Resolution MEPC.320(74) [4], adopted on 17 May 2019, established the guidelines for ensuring the consistent implementation of the 0.5% sulphur limit under MARPOL Annex VI. These guidelines established a set of recommendations and technical considerations, as well as several requirements to be used by shipowners, administrations, shipbuilders and fuel oil suppliers regarding fuel oil availability, fuel oil tanks and fuel oil system configuration, among others. The above-mentioned guidelines also set out several recommendations to be followed by Port State Control Officers during inspections. In this context, Resolution MEPC.321(74) [5], adopted on 17 May 2019, establishes the specific guidelines for Port State Control Officers performing inspections under MARPOL Annex VI.

In this scenario, the current and available options that shipping companies have for following this specific regulation are very limited. One of them is the use of a compliant fuel whose content is less than or equal to 0.5% mass by mass. During our research, a tracking of ships calling into Barcelona port during the period between January 2020 and

January 2022 was carried out in terms of if they are using very low sulphur fuel oil (VLSFO) or an exhaust-gas cleaning system (EGCS) during their calls in port. After analysing all the obtained data, we can assert that the use of VLSFO is currently the most-chosen option among ships calling Barcelona port. The main problem with this option is that the ship depends on the availability of this fuel oil and its price in each port is different. Currently (January 2022), the price of very low sulphur fuel oil in a European port such as Rotterdam has peaked at 652.5 \$/mt as showed in Figure 1. From 20 December 2021, the price of VLSFO (529 \$/mt) has experienced, in general terms, a very strong increasing trend that seems to be likely to continue in the future.

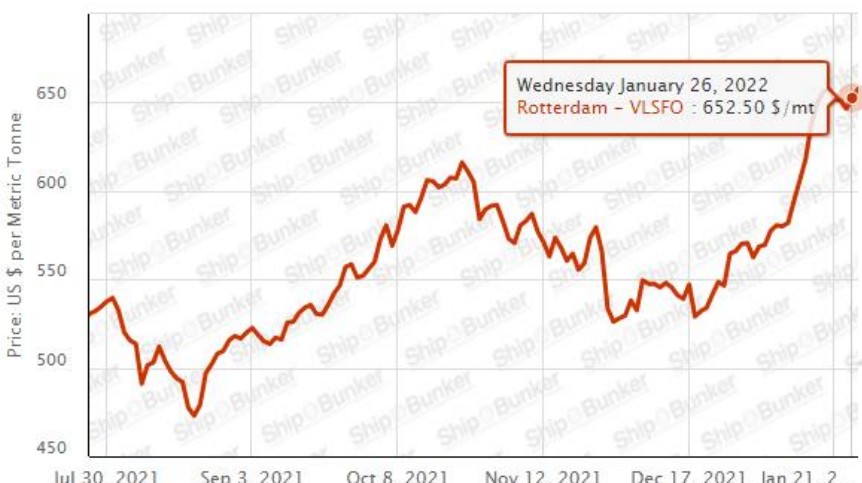

**Figure 1.** View of price evolution for VLSFO. Source: shipandbunkers.com (accessed on 14 March 2022).

Another option, one less chosen by shipping companies in the case of existing ships, is to use alternative clean fuels—for instance, natural gas. The installation of natural gas propulsion onboard an existing ship requires a great investment, and the lack of sufficient space onboard usually requires a long time in a dry dock. Ship owners mainly use this option for new ships. In Barcelona port, two ships engaged in a regular service between the Balearic Islands and Barcelona that call into the port every day are fitted with liquified natural gas engines; one of them is the Ro/Pax ship "ABEL MATUTES" from the company "BALEARIA". This ship was refitted with dual-fuel engines that can use natural gas and conventional fuels. Furthermore, the ship is fitted with two natural gas storage tanks of 178 m$^3$ each (Figure 2).

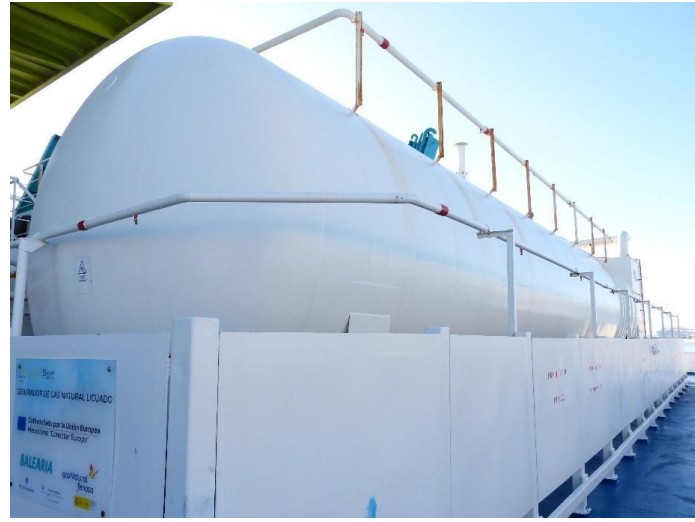

**Figure 2.** View of LNG storage tank onboard the ship "ABEL MATUTES". Source: The authors.

Finally, the third most-chosen option by shipping companies for dealing with this new regulation is to install exhaust-gas cleaning systems onboard their ships, also known as "scrubbers".

These systems represent, as stated in research regarding the costs–benefits of strategies in compliance with low sulphur [6], a great investment for shipping companies; furthermore, the profitability of this investment is strongly linked to the ship's life. As per the Bureau Veritas publication (2020) [7], the average cost of an exhaust-gas cleaning system installation is about 2 million dollars per scrubber; depending on the vessel and depending on fuel oil evolution prices, the return on investment (ROI) period for an exhaust-gas cleaning system installation is between 1–4 years [6]. It seems to make no sense to invest in this for a ship whose useful life is going to expire soon.

After the use of a compliant fuel, the third option (use of scrubbers) is what most ship owners are choosing for their ships. These systems are classified generally into two main groups, as follows: dry scrubbers and wet scrubbers. Dry scrubbers, basically, remove sulphur oxides from exhaust gases through the action of granules of hydrated lime. During the exhaust-gas cleaning process, dry scrubbers do not generate wastewater to discharge into the sea.

Wet scrubbers use water that is sprayed over the exhaust-gas cleaning system during the cleaning process to remove sulphur oxides. Wet scrubbers are also classified into three main groups: open-loop systems, closed-loop systems and hybrids.

This paper is focused on the research we are carrying out on the effects produced by the discharge of residual wash water from open-loop exhaust-gas cleaning systems into port water. We are studying the pH level of Barcelona port water and its changes, as well as its own capacity to neutralize the acidic waters from scrubber discharges.

Currently there are several studies regarding the effect that wash water discharges from open-loop scrubbers are having on seawater and/or port water parameters; we should highlight some of them, based on the aim of the research and the results obtained. All scrubbers—open-loop, closed-loop and hybrid—discharge water that is more acidic and turbid than the surrounding water. This contributes to ocean acidification and worsens water quality (Bryan Comer, Ph.D., Elise Georgeff and Liudmila Osipova, Ph.D. (2020)) [8]. The use of these scrubbers allows ships to run on heavy fuel and is potentially responsible for a general decrease in pH of at least 0.004 over the whole area of study, including the English Channel and the southern part of the North Sea, and to a decrease of 0.031 or more in coastal areas with intense shipping traffic (Dulière V., Baetens K. and Lacroix G. (2020)) [9]. The most popular type of scrubber, open loop, constantly discharges large amounts of wash water that is acidic and contains polycyclic aromatic hydrocarbons (PAHs), particulate matter, nitrates, nitrites and heavy metals including nickel, lead, copper and mercury—all of which are discharged into the aquatic environment where they can damage marine ecosystems and wildlife and worsen water quality (Liudmila Osipova, Ph.D., Elise Georgeff and Bryan Comer, Ph.D. (2021)) [10]. The average difference between inlet and outlet water pH was 2.05, meaning that on average, the outlet water was 102.05 (=110)-times more acidic than the inlet water (Anna Lunde Hermansson, Ida-Maja Hassellov, Jana Moldanova, Erik Ytreberg (2021)) [11].

This paper, as well as giving a picture of how the pH level of South Basin Barcelona port water is evolving with the use of open-loop scrubbers discharging their acidic residual wash water, also presents an alternative for avoiding these systems that contributes to acidification—thus improving their operation.

The first part of this paper talks about the current legal framework that regulates the use of exhaust-gas cleaning systems. In this section, we show a general overview of the main regulations and requirements that scrubbers must meet.

Secondly, we analyse the current situation, considering the last research on this topic, studying the availability of new technologies applied to this problem and making a compilation of main ideas from several scientific papers on this subject.

Thirdly, we explain the methodology that we have followed in this research, with an analysis of the data obtained and a study of the trends for certain water parameters.

Afterwards, we provide a summary of the preliminary results obtained, analysing the current scenario of Barcelona port water parameters and their possible evolution.

Finally, we discuss the conclusions based on the achieved results, introducing new measures for improving the operation of exhaust-gas cleaning systems.

## 2. Legal Framework

Regulation 14 [1]—sulphur oxides and particulate matter, of Annex VI [3]—Prevention of Air Pollution from Ships—MARPOL Convention [2], makes the implementation of sulphur limits on the fuels used for combustion purposes onboard ships mandatory.

Regulation 4.1 [12] of the same Annex, regarding equivalent arrangements used onboard ships in order to be in compliance with regulation 14, establishes that "The Administration of a Party may allow any fitting, material, appliance or apparatus to be fitted in a ship or other procedures, alternative fuel oils, or compliance methods used as an alternative to that required by this Annex if such fitting, material, appliance or apparatus or other procedures, alternative fuel oils, or compliance methods are at least as effective in terms of emissions reductions as that required by this Annex, including any of the standards set forth in regulations 13 and 14".

From the above, it is clear that exhaust-gas cleaning systems are effectively considered equivalent arrangements.

In addition to the previous regulation, the Resolution from the Marine Environment Protection Committee MEPC.259(68) [13] determines in Annex 10, among other things, the wash-water discharge criteria. This point states that the wash water used in the systems for the exhaust-gas cleaning process must be continuously monitored and recorded when this system is being used by ships in ports, harbours or estuaries.

The values which must be monitored are, at minimum, pH, PAH (Polycyclic Aromatic Hydrocarbons), turbidity/suspended particulate matter and temperature. For the parameters listed above, this Resolution indicates the limits which any wash-water discharge from scrubbers must comply with.

Furthermore, Resolution MEPC.259(68) [13] defines both schemes, A and B, for the approval, survey and certification of an exhaust-gas cleaning system and establishes the minimum content and all requirements for Technical Manuals for the above-mentioned schemes.

MEPC.259(68) [13] also determines the type and scope of inspection that each exhaust-gas cleaning system is subjected to, as well as the documentation set that a ship fitted with a scrubber system must have available onboard for inspection if required by authorities—for instance, during Port State Control. This documentation is set out in the above-mentioned Resolution and, depending on the scheme, is required to be as follows:

Ships fitted with an exhaust-gas cleaning system certified as per Scheme A must have onboard [9]:

1.  Sulphur Oxides Emissions Compliance Plan (SECP) approved by the Administration.
2.  Sulphur Oxides Emissions Compliance Certificate.
3.  Technical Manual for Scheme A approved by the Administration.
4.  Onboard Monitoring Manual approved by the Administration.
5.  EGC Record Book, with a form approved by the Administration.

Ships fitted with an exhaust-gas cleaning system certified as per Scheme B must be provided with [9]:

1.  Sulphur Oxides Emissions Compliance Plan (SECP) approved by the Administration.
2.  Technical Manual for Scheme B approved by the Administration.
3.  Onboard Monitoring Manual approved by the Administration.
4.  EGC Record Book, with a form approved by the the Administration.

### 3. Current Situation

As we have seen before, one of the options that ship owners have to follow is Regulation 14, Annex VI, MARPOL Convention, which is to install an exhaust-gas cleaning system (scrubbers) onboard their ships. According to the DNV GL study [14], carried out in 2018 regarding the use of scrubbers for dealing with the new sulphur cap, which entered into force on 1 January 2020, around 80% of scrubbers installed onboard ships are in open-loop mode.

In the specific case of ships calling into Barcelona port with a scrubber installed onboard, as a result of the above-mentioned specific tracking performed for ships that have called into Barcelona port between January 2020 and January 2022, approximately 69% of them had the hybrid option to discharge wash water into the sea and 30% were directly open loop (continuous discharges). Therefore, only 1% of the ships calling into Barcelona port with a scrubber installed onboard were closed loop.

The main characteristic of the open-loop scrubbers is that the seawater used during the exhaust-gas cleaning process (residual wash water) is directly discharged back into the sea. The discharged wash water must follow certain water parameters (wash water criteria) established by IMO Resolution MEPC.259(68) [13] "2015 guidelines for exhaust-gas cleaning systems". Among these water parameters regulated by the IMO Resolution is the pH, which is the focus of our research.

Nowadays, there are several studies around the world looking into the impact of open-loop scrubber discharges. Most of them are focused on the effect that wash water discharges have on the marine environment, specifically on port water, harbour docks, estuaries [15] and coastal water [16]. In this paper, the acidic component of wash water discharged from the open-loop exhaust cleaning system is revealed, among other features. As stated in the paper, the exhaust gases—which are conducted through an exhaust-gas cleaning system scrubbing tower that is sprayed with seawater—produces, due to the high content of sulphur oxides in these exhaust gases, sulphurous acid ($H_2SO_3$) and sulphuric acid ($H_2SO_4$). This research was developed on the effects of wash water discharges from ships into the Antwerp harbour docks and the Scheldt estuaries (Belgium). In these scenarios, the study of the evolution of pH harbour docks and Scheldt estuaries was carried out, considering that 20% of ships calling into these docks are using open-loop scrubbers. The acidic sulphur compounds that are discharged directly into the surface water were observed and their acidifying capacity was greater than vessels operating on low-sulphur fuel (0.1% m/m). In fact, this research showed a decrease in pH of 0.015 units, directly caused by wash water discharge.

Previous research conducted in the year 2015 [12] was focused on the effects of wash water discharges from scrubbers into the North Sea and Baltic Sea, in German coastal waters. This research was focused mainly on the effects on the marine biota of these regions and determined the changes in pH values caused by wash water discharges from scrubbers. It was confirmed that these changes occurred on a small scale and that the acidification appeared faster in limited or closed systems than in the case of the North Sea and Baltic Sea.

This acidification, as concluded in this research, will first affect the phytoplankton, and then any changes in these organisms will affect other biological components:

"The direct impacts of lowering the pH on marine mammals and birds are not serious, but the impact on phytoplankton and thus on the whole food web will also become evident for higher organisms."

"A pH value reduction will have a greater effect in closed water bodies than on open sea."

Some other research regarding the buffering capacity of seawater and oceans has been worked on by authors [17], concluding that the ocean buffer capacity is being reduced due to increasing atmospheric $CO_2$ levels. According to this paper, the seawater buffer capacity will decrease due to future atmospheric $CO_2$ levels by, on average, 34% from 2000 to 2100;

additional research [18] has determined that this kind of activity may contribute to seawater buffering via dissolution, precipitation and mineral surface reactions.

In sight of the above-mentioned papers and their conclusions, we can assume that the wash-water discharges from open-loop exhaust gas-cleaning systems clearly have an acidic component and that it is changing, at least on very small scale, the seawater pH value. These small changes in water pH could endanger the marine biota, first starting to affect phytoplankton. Furthermore, these wash-water discharges are contributing to an ocean buffer capacity reduction due to their acidic component.

## 4. Data Analysis

### 4.1. Introduction

At present, and since the beginning of 2021, we are analysing the evolution of the acidity from Barcelona port water. We have chosen a specific area of the Barcelona port: the South basin. As a result of our research, we have observed that in this port area there are currently berthing ships that are using exhaust-gas cleaning systems during their calls to port—most of them operating in open-loop mode. This is the reason we have chosen this area for our study.

Inside the South Basin of Barcelona port there are mainly car carrier ships berthing. As mentioned before, most of them use open-loop exhaust-gas cleaning systems during their calls. We have to consider that there are two main scenarios regarding the use of open-loop scrubbers onboard these car carriers inside the South Basin Barcelona port; on the one hand is the moment that the ship is manoeuvring for berthing; on the other hand, is when a ship is manoeuvring on departure from the quay. In both situations, the ship is using its main and auxiliary engines at a substantial engine load (%) and the SOx emissions are greater in these moments and, consequently, more acidic water is formed.

The average car carrier ship berthing in the South Basin Barcelona port has the following characteristics:

- Type of ship: car carrier
- Main dimensions: 167 m (length) × 28 m (beam)
- Main engine power: 1 × 11,000 kW
- Auxiliary engines: 4 × 1400 kW
- Wash water flow average: 300–350 m$^3$/h

Furthermore, we must consider the particular situation of this area inside the Barcelona port, which is quite enclosed and not exposed to the open sea at all (Figures 3 and 4).

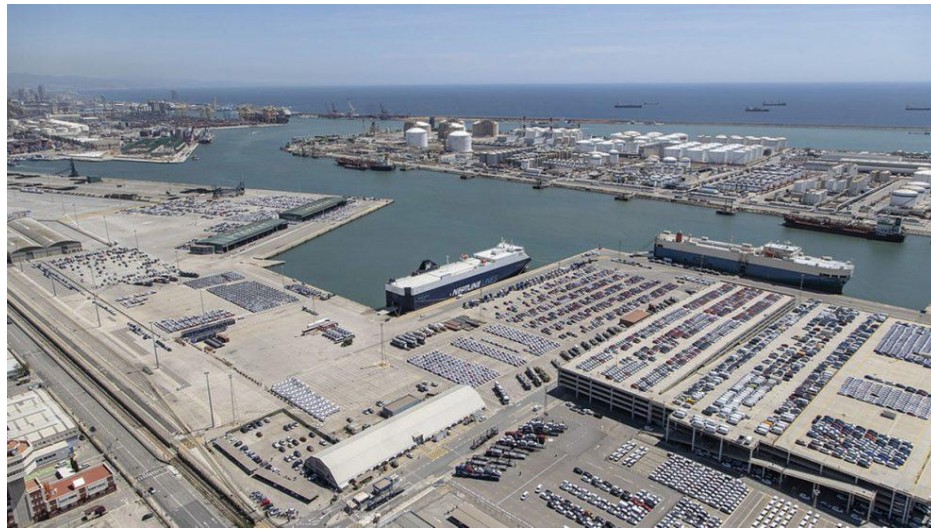

**Figure 3.** View of the South basin of Barcelona port. Source: elmercantil.com, (accessed on 14 March 2022).

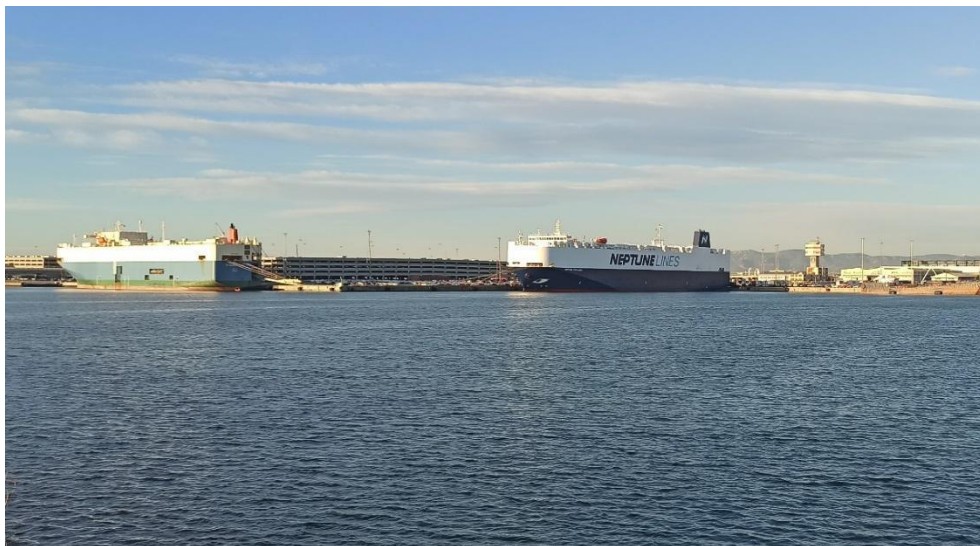

**Figure 4.** View of the South basin of Barcelona port. Source: The Authors.

In order to carry out the analysis of port water, a collaboration with an independent laboratory—the Institute of Environmental Assessment and Water Research (IDAEA), which is part of the Superior Council of Scientific Research (CSIC)—has been carried out for the samples taken from South basin of Barcelona port, determining its pH level. We have been analysing up to three samples per month through the IDAEA laboratory during the first stage.

Furthermore, we have been carrying out the sampling and pH analysis of the South basin water two–three times per week through a digital device approved for measuring pH. The results obtained from this equipment have been compared and complemented with the results obtained from the Institute of Environmental Assessment and Water Research (IDAEA).

*4.2. IDAEA pH Data*

We started the sampling and pH analysis of Barcelona port South basin water with the Institute of Environmental Assessment and Water Research (IDAEA-CSIC) at the beginning of the year 2021 (Figure 5). After carefully studying the first data obtained from these analyses, we have observed that, initially, there was a clear decreasing trend in the South basin water pH levels. We are working with very small values regarding the pH changes, but the aim of this investigation lies in the early detection of a long-term change.

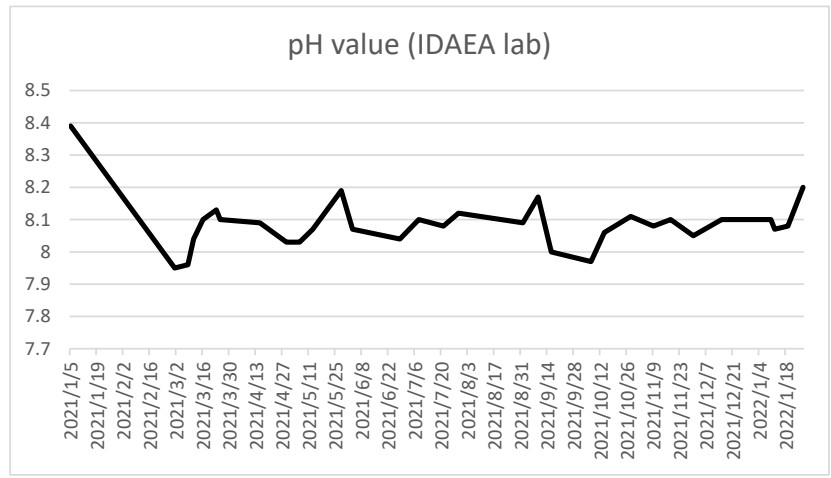

**Figure 5.** IDAEA pH results from the South basin Barcelona port water. Source: The authors.

The above-mentioned decreasing trend in the pH values continued until the first days of March, when they stabilised with a slight increasing trend.

We have also noted as that the results of the pH analysis during the second and third week of March 2021 showed a strong upward trend. From this moment, and during the month of April, we were able observe a slight decreasing tendency until the second week of May, when the pH level increased, exceeding the value of 8.18 units. From this moment there was a strong downward trend that continued until the third week of June, when it started to increase, reaching 8.12 pH units. The situation of the pH level was quite constant until 9 September 2021, when the pH level reached the value of 8.17 units. At the last analysis (27 January 2022), after several small changes, the pH level of the Barcelona port water (South Basin) was 8.2 pH units—the second highest value obtained so far from the CSIC laboratory analysis.

All these variations in the pH level were mainly due to the frequency of ships calling into the South Basin of the Barcelona port using open-loop exhaust-gas cleaning systems during the period of sampling, although they were also to a lesser extent due to the intensity, duration and frequency of precipitation in this area.

### 4.3. pH Meter and Refractometer Data

The sampling and analyses performed with the approved pH meter started at the beginning of March 2021 (Figure 6).

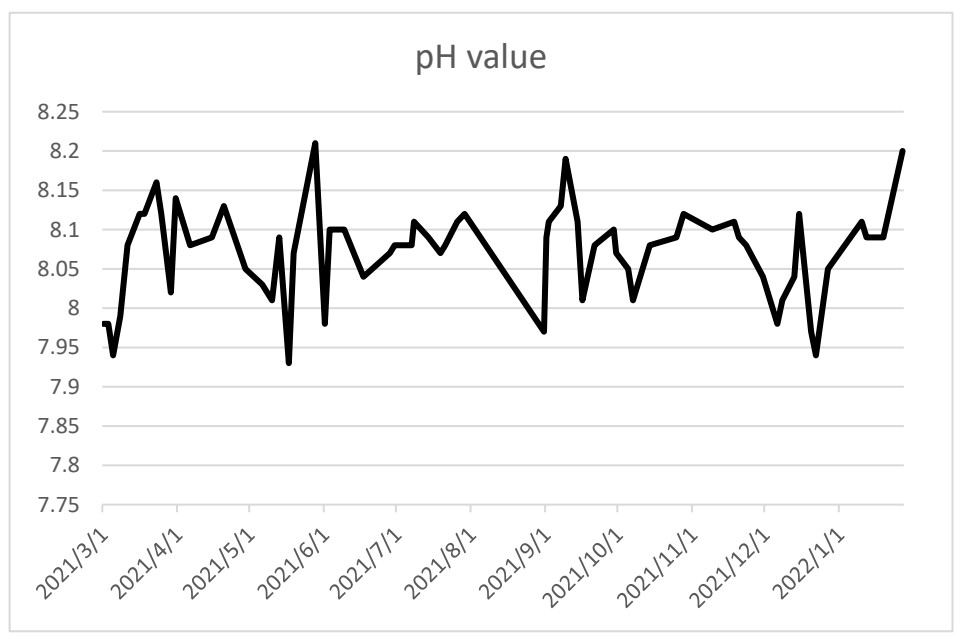

**Figure 6.** pH-meter results from the South Basin Barcelona port water. Source: The authors.

All the results obtained were really close to, and in several cases coincident with, the pH results from the IDAEA laboratory. In this way, we can assert that the results obtained from the pH meter were frequently corroborated by those obtained from the Institute of Environmental Assessment and Water Research laboratory (IDAEA-CSIC).

All the data obtained through the pH meter shows in detail how the pH level of the South basin water has been changing since the beginning of March—the moment in which the downward-trend in pH levels that had occurred since the beginning of year stabilised and began an upward trend. The use of this system helps us to see in more detail—since the pH of the water is analysed more frequently—how the pH changes from week to week.

Furthermore, we have been analysing, through the above-mentioned equipment and using a refractometer as well, other water parameters causally related to the port water basicity. We have analysed the the Oxidation-Reduction Potential (mV) (Figure 7), the salt concentration (g/L) (Figure 8) and the water density (kg/L) (Figure 9).

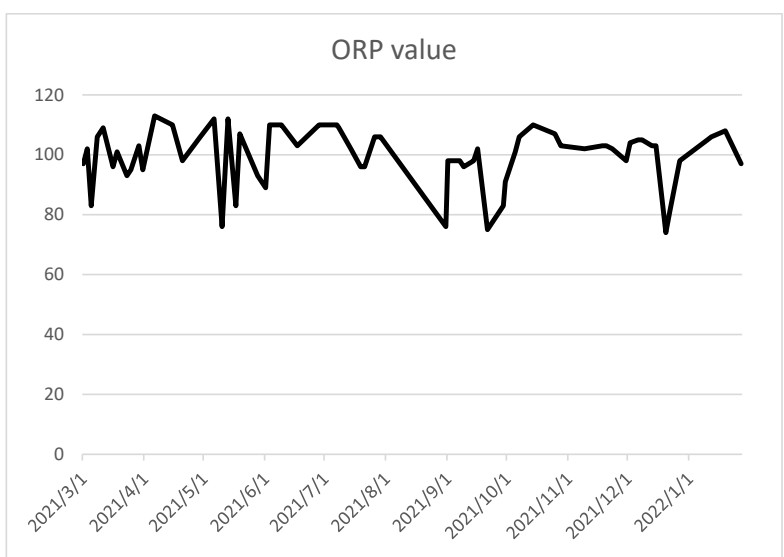

**Figure 7.** Oxidation-Reduction Potential results from the South Basin Barcelona port water. Source: The authors.

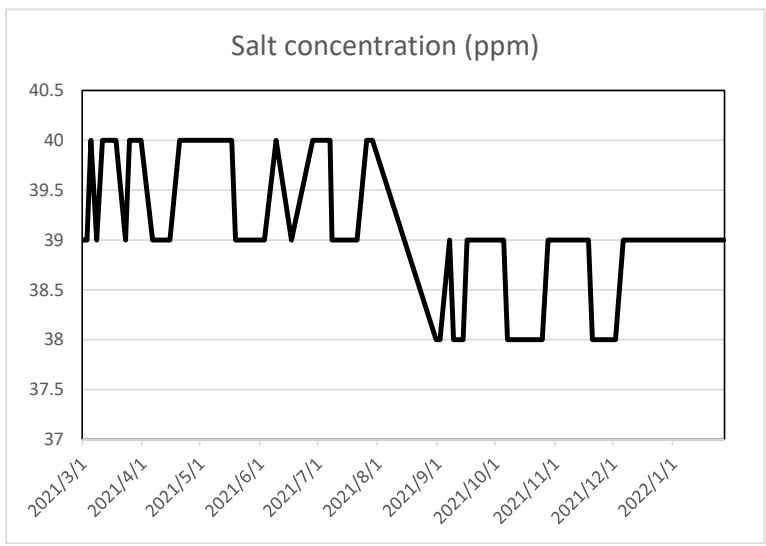

**Figure 8.** Salt concentration results from the South basin Barcelona port water. Source: The authors.

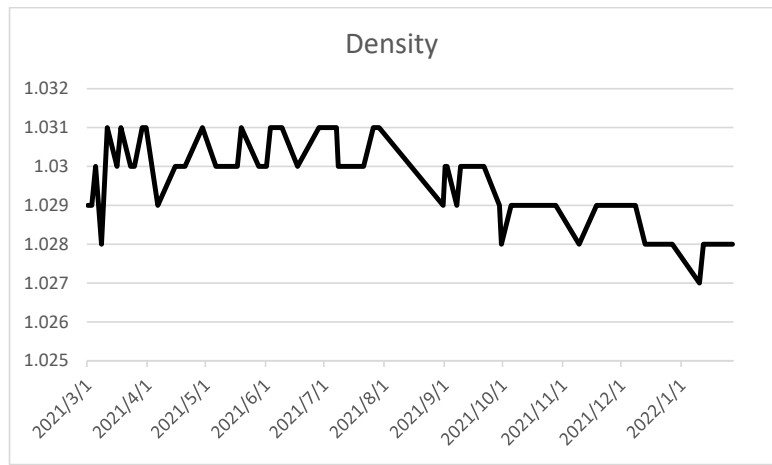

**Figure 9.** Diagram of density results from the South basin Barcelona port water. Source: The authors.

The highest pH value measured with this equipment was observed on 28 May 2021, at 8.21. Then, the pH level showed a strong downward trend at the beginning of June, until it stabilised at the last analysis on 9 June 2021 with a pH value of 8.1 units. It is at the end of July when the pH level experiments showed a clear downward trend until 31 August 2021, when it reached their third lowest value, 7.97 pH units. From this moment, we have observed a strong upward trend up to 8.19 pH units (9 September 2021). Then, the pH level showed a weaker decreasing trend until 14 October 2021, with a pH level of 8.08 units.

From this date, the pH level experienced several small increases and decreases. In the analysis carried out on 22 December 2021, the pH value was found to be 7.94 units. These continuous and slight variations may be due to the small rainfalls that took place in mid-December in Barcelona.

The last analysis performed on 27 January 2022 showed an increasing trend in port water pH, reaching the second highest value obtained so far with this specific equipment (8.2 pH units).

As mentioned before, for the case of pH levels, variations observed in the Oxidation-Reduction Potential value were mainly due to the frequency of ships calling into the South Basin of the Barcelona port using open-loop scrubbers, and due to the frequency of precipitation.

In the case of the salt concentration and port water density, all the variations observed were mainly due to the frequency, intensity and duration of precipitation, although also to a lesser extent by the frequency of ships calling into the South Basin of the Barcelona port using open-loop exhaust-gas cleaning systems.

## 5. Preliminary Results

An extract of the results obtained (Table 1) regarding the South Basin port seawater pH value from both the CSIC laboratory and our specific equipment are shown as follows.

**Table 1.** Extract of results obtained regarding the South Basin port seawater pH values from 5 January 2021 to 20 April 2021.

| Date of Analysis | Place | pH Value (CSIC) | pH Value (Specific Equipment) | ORP Value (Specific Equipment) | Density | Salt Concentration (ppm/%; Refractometer) |
|---|---|---|---|---|---|---|
| 05/01/2021 | South Basin | 8.39 | N/A | N/A | 1.029 | N/A |
| 01/03/2021 | South Basin | 7.95 | 7.98 | 97 mV(+) | 1.029 | 39,000 ppm (3.9%; 39 g/L) |
| 03/03/2021 | South Basin | N/A | 7.98 | 102 mV(+) | 1.03 | 39,000 ppm (3.9%; 39 g/L) |
| 05/03/2021 | South Basin | N/A | 7.94 | 83(+) | 1.028 | 40,000 ppm (4%; 40 g/L) |
| 08/03/2021 | South Basin | 7.96 | 7.99 | 106(+) | 1.031 | 39,000 ppm (3.9%; 39 g/L) |
| 11/03/2021 | South Basin | 8.04 | 8.08 | 109(+) | 1.03 | 40,000 ppm (4%; 40 g/L) |
| 16/03/2021 | South Basin | 8.1 | 8.12 | 96(+) | 1.031 | 40,000 ppm (4%; 40 g/L) |
| 23/03/2021 | South Basin | 8.13 | 8.16 | 93(+) | 1.03 | 39,000 ppm (3.9%; 39 g/L) |
| 25/03/2021 | South Basin | 8.1 | 8.12 | 95(+) | 1.03 | 40,000 ppm (4%; 40 g/L) |
| 29/03/2021 | South Basin | N/A | 8.02 | 103(+) | 1.031 | 40,000 ppm (4%; 40 g/L) |
| 31/03/2021 | South Basin | N/A | 8.14 | 94(+) | 1.031 | 40,000 ppm (4%; 40 g/L) |
| 06/04/2021 | South Basin | N/A | 8.08 | 113(+) | 1.029 | 39,000 ppm (3,9%; 39 g/L) |
| 15/04/2021 | South Basin | 8,09 | 8.09 | 110 | 1.03 | 39,000 ppm (3,9%; 39 g/L) |
| 20/04/2021 | South Basin | N/A | 8.13 | 98 | 1.03 | 40,000 ppm (4%; 40 g/L) |

A complete table of results obtained on the South Basin port seawater pH values from both the CSIC laboratory and our specific equipment is attached in Appendix A.

After the analysis of the obtained data, we have observed the following extremes:

The pH level recovery of South basin water is, as far as we can ascertain for the time being, very high. This is related to the basicity or alkalinity of this water, a concept that

determines the capacity of solutes in aqueous systems to neutralize acid. We can observe that there was a downward trend in pH levels from the beginning of the year to the end of February. This downward trend was coincident with several ships calling during this period into the South basin. Most of these ships were using open-loop exhaust-gas cleaning systems during their calls. However, this decreasing trend was countered, stabilised and even modified upward in the short term. In fact, this pH level recovery even took place with some other ships calling into the South basin Barcelona port using exhaust-gas cleaning systems in an open-loop mode.

We can observe that the pH level in every period was causally related with the values of some parameters such as the salt concentration and density of water, among others. This is because the salt concentration of water is a clear indication of the basicity level of the water. As stated by the Arrhenius definition of acids and bases, an acid is a substance such as sulphuric acid that dissolves in water to produce H+ ions (also known as protons) and a base is a substance such as sodium hydroxide that dissolves in water to produce hydroxide ions (OH-; Chemistry libretext website) [19]. In this way, we must remember that water is an ampholyte and that it can behave like an acid or a base.

For the above-mentioned scenario, and considering our own scenario, the South basin water of the Barcelona port, depending on certain external factors such wash-water discharges from open-loop exhaust-gas cleaning systems that have an acidic component such as sulphuric acid, port water can behave like an acidic solution. However, the South basin Barcelona port water may have an extremely high alkalinity due to its high concentration of bicarbonates ($HCO_{3-}$) and carbonates ($CO_{3-2}$).

The latter would explain the great capacity to resist change in pH that the South basin Barcelona port water has. This capacity is defined as the buffering capacity of the water. This is because the above-mentioned bicarbonates and carbonates react with the released H+ from an acid, such as sulphuric acid dissociating in water, preventing the port water pH from dropping.

In all of what is mentioned above lies the importance of this investigation, checking the current situation of the pH levels of the South basin Barcelona port water and its short and long-term trends after having continuous discharges of wash water from scrubbers installed onboard the ships that are operating in open-loop mode during their calls at Barcelona port.

The sampling and analysis of the South basin Barcelona port water pH has been carried out and studied for approximately 2 years and the obtained results are being studied and introduced into a database showing the current situation of the pH levels and their short and long-term trends.

## 6. Discussion

Considering all of what is mentioned above, and despite there currently being no immediate need to reduce the acidity level of the South basin Barcelona port water after its high basicity has been ascertained, we are investigating to find a way to improve open-loop exhaust-gas cleaning systems—in particular, a way to reduce the acidity of the wash waters used during the exhaust-gas cleaning process that are discharged into the sea.

In our research we are going to propose a preventive technique that is to be applied directly to the exhaust-gas cleaning system treatment plants, where, through the implementation of a new step in the wash water treatment, this water is going to be discharged into the sea with almost the same pH as when it was collected. In this way, the current pH of the wash waters discharged from open-loop exhaust-gas cleaning systems, which is around 4–5 pH units measured on the overboard discharge point, will be increased to 7.5–8.2 pH units.

The system proposed is based on the neutralisation reaction, controlling the acidity of residual wash water. Furthermore, to prevent the generation of new residues or increases in salt concentration, an additional treatment will be performed to the wash water before is discharged.

Currently, we expect to start the testing of this proposed system on a small scale soon. Despite this, it is too early to talk about results in terms of the efficiency with which we hope to contribute to seawater acidification reduction, improving the operation of open-loop exhaust-gas cleaning systems.

## 7. Conclusions

That wash water from exhaust-gas cleaning systems has a strong acidic component is a fact. The exhaust-gas cleaning systems' manufacturer manuals consulted confirm this, as they allow a certain pH level in the wash water discharges.

The results obtained so far on the South basin port water pH values confirms that Barcelona port water has a great capacity to assume changes in pH. We have mostly noticed that, generally, the South Basin port water pH recovers to its normal values in a few days after experiencing a small pH drop. We can check the latest data from 17 May 2021, when we observed the lowest South basin port water pH value of 7.93 units. On 19 May 2021, we analysed a new South basin port water sample that was 8.07 pH units, and on 28 May 2021, a new sample was analysed with a value of 8.21 pH units. The latter defines the great capacity of the South Basin Barcelona port water for neutralizing the acidity, in this case, of wash water from scrubbers.

Keeping in mind the-above mentioned research, we can assert that seawater's buffer capacity is not unlimited. Considering that open-loop exhaust-gas cleaning systems also contribute to the acidification of seawater with wash water discharges, at the same time reducing the seawater buffer capacity, we can conclude that these wash-water discharges accelerate the process by which the seawater buffering limit is achieved.

The alkalinity of water, in general terms, is the capacity of water to deal with changes in pH—the resistance of water to becoming acidic. It is also defined as the buffering capacity of the water. This buffering capacity, in the case of seawater, is determined by the presence of certain components such as soluble hydroxide, carbonates and bicarbonates. The higher the alkalinity of the seawater, the higher is its capacity to resist changes in pH. Alkalinity is usually measured in terms of Calcium Carbonate ($CaCO_3$ ppm).

In a hypothetical case where all the buffering components in the South Basin Barcelona port water are neutralized by an acid, the concentration of H+ ions (hydrogen ions, also known as protons) in the water would be increased quickly, with a consequent drastic drop in seawater pH (acidification).

To prevent this hypothetical situation, our investigation is focused on improving, as far as possible, the operation of open-loop exhaust-gas cleaning systems by introducing a new seawater pre-treatment phase before its being used in the scrubbing process.

This new phase will allow ships using open-loop scrubbers to discharge wash water with a pH as neutral as possible into the marine ecosystem, almost at the same level as when the seawater was taken in by the system's inlet.

Our project is based on the introduction of an automatic system that continuously receives the pH data of water monitored at the outlet rack. Once the system has this data, it will compare it with the pH measurement at the inlet rack. In case a drop of pH is detected between the inlet and outlet water, the system will automatically provide an extra alkaline mixture to the tank for use during the exhaust-gas cleaning process at the scrubbing tower.

**Author Contributions:** Conceptualization, N.G.D.D. and F.X.M.D.O.; methodology, F.X.M.D.O.; validation, N.G.D.D. and F.X.M.D.O.; formal analysis, N.G.D.D.; investigation, N.G.D.D.; resources, F.X.M.D.O.; data curation, N.G.D.D.; writing—original draft preparation, N.G.D.D.; writing—review and editing, F.X.M.D.O.; supervision, F.X.M.D.O. All authors have read and agreed to the published version of the manuscript.

**Funding:** This research received no external funding and was only performed with department funds.

**Data Availability Statement:** The data presented in this study are available on request from the corresponding author.

**Conflicts of Interest:** The authors declare no conflict of interest.

## Appendix A. Results Obtained on the South Basin Port Seawater pH Value from Both the CSIC Laboratory and Our Specific Equipment

| Date of Analysis | Place | pH Value (CSIC) | pH Value (Specific Equipment) | ORP Value (Specific Equipment) | Density | Salt Concentration (ppm/%; Refractometer) |
|---|---|---|---|---|---|---|
| 05/01/2021 | South Basin | 8.39 | N/A | N/A | 1.029 | N/A |
| 01/03/2021 | South Basin | 7.95 | 7.98 | 97 mV(+) | 1.029 | 39,000 ppm (3.9%; 39 g/L) |
| 03/03/2021 | South Basin | N/A | 7.98 | 102 mV(+) | 1.03 | 39,000 ppm (3.9%; 39 g/L) |
| 05/03/2021 | South Basin | N/A | 7.94 | 83(+) | 1.028 | 40,000 ppm (4%; 40 g/L) |
| 08/03/2021 | South Basin | 7.96 | 7.99 | 106(+) | 1.031 | 39,000 ppm (3.9%; 39 g/L) |
| 11/03/2021 | South Basin | 8.04 | 8.08 | 109(+) | 1.03 | 40,000 ppm (4%; 40 g/L) |
| 16/03/2021 | South Basin | 8.1 | 8.12 | 96(+) | 1.031 | 40,000 ppm (4%; 40 g/L) |
| 23/03/2021 | South Basin | 8.13 | 8.16 | 93(+) | 1.03 | 39,000 ppm (3.9%; 39 g/L) |
| 25/03/2021 | South Basin | 8.1 | 8.12 | 95(+) | 1.03 | 40,000 ppm (4%; 40 g/L) |
| 29/03/2021 | South Basin | N/A | 8.02 | 103(+) | 1.031 | 40,000 ppm (4%; 40 g/L) |
| 31/03/2021 | South Basin | N/A | 8.14 | 94(+) | 1.031 | 40,000 ppm (4%; 40 g/L) |
| 06/04/2021 | South Basin | N/A | 8.08 | 113(+) | 1.029 | 39,000 ppm (3.9%; 39 g/L) |
| 15/04/2021 | South Basin | 8.09 | 8.09 | 110 | 1.03 | 39,000 ppm (3.9%; 39 g/L) |
| 20/04/2021 | South Basin | N/A | 8.13 | 98 | 1.03 | 40,000 ppm (4%; 40 g/L) |
| 29/04/2021 | South Basin | 8.03 | 8.05 | 106 | 1.031 | 40,000 ppm (4%; 40 g/L) |
| 06/05/2021 | South Basin | 8.03 | 8.06 | 105 | 1.03 | 40,000 ppm (4%; 40 g/L) |
| 10/05/2021 | South Basin | N/A | 8.01 | 76 | 1.03 | 40,000 ppm (4%; 40 g/L) |
| 13/05/2021 | South Basin | 8.07 | 8.09 | 112 | 1.03 | 40,000 ppm (4%; 40 g/L) |
| 17/05/2021 | South Basin | N/A | 7.93 | 83 | 1.03 | 40,000 ppm (4%; 40 g/L) |
| 19/05/2021 | South Basin | N/A | 8.07 | 107 | 1.031 | 39,000 ppm (3.9%; 39 g/L) |
| 28/05/2021 | South Basin | 8.19 | 8.21 | 93 | 1.03 | 39,000 ppm (3.9%; 39 g/L) |
| 01/06/2021 | South Basin | N/A | 7.98 | 89 | 1.03 | 39,000 ppm (3.9%; 39 g/L) |
| 03/06/2021 | South Basin | 8.07 | 8.1 | 110 | 1.031 | 39,000 ppm (3.9%; 39 g/L) |
| 09/06/2021 | South Basin | N/A | 8.1 | 110 | 1.031 | 40,000 ppm (4%; 40 g/L) |
| 28/06/2021 | South Basin | 8.04 | 8.07 | 110 | 1.03 | 40,000 ppm (4%; 40 g/L) |
| 30/06/2021 | South Basin | N/A | 8.08 | 110 | 1.031 | 40,000 ppm (4%; 40 g/L) |
| 07/07/2021 | South Basin | N/A | 8.08 | 110 | 1.031 | 40,000 ppm (4%; 40 g/L) |
| 08/07/2021 | South Basin | 8.1 | 8.11 | 109 | 1.03 | 39,000 ppm (3.9%; 39 g/L) |
| 14/07/2021 | South Basin | N/A | 8.09 | 102 | 1.03 | 39,000 ppm (3.9%; 39 g/L) |
| 19/07/2021 | South Basin | N/A | 8.07 | 96 | 1.03 | 39,000 ppm (3.9%; 39 g/L) |
| 21/07/2021 | South Basin | 8.08 | 8.08 | 96 | 1.03 | 39,000 ppm (3.9%; 39 g/L) |
| 26/07/2021 | South Basin | N/A | 8.11 | 106 | 1.031 | 40,000 ppm (4%; 40 g/L) |
| 29/07/2021 | South Basin | 8.12 | 8.12 | 106 | 1.031 | 40,000 ppm (4%; 40 g/L) |
| 31/08/2021 | South Basin | N/A | 7.97 | 76 | 1.029 | 38,000 ppm (3.8%; 38 g/L) |
| 01/09/2021 | South Basin | 8.09 | 8.1 | 98 | 1.03 | 38,000 ppm (3.8%; 38 g/L) |
| 02/09/2021 | South Basin | N/A | 8.11 | 98 | 1.03 | 38,000 ppm (3.8%; 38 g/L) |
| 07/09/2021 | South Basin | N/A | 8.13 | 98 | 1.029 | 39,000 ppm (3.9%; 39 g/L) |
| 09/09/2021 | South Basin | 8.17 | 8.19 | 96 | 1.03 | 38,000 ppm (3.8%; 38 g/L) |
| 14/09/2021 | South Basin | N/A | 8.11 | 98 | 1.03 | 38,000 ppm (3.8%; 38 g/L) |
| 16/09/2021 | South Basin | 8 | 8.01 | 102 | 1.03 | 39,000 ppm (3.9%; 39 g/L) |

| 21/09/2021 | South Basin | N/A | 8.08 | 75 | 1.03 | 39,000 ppm (3.9%; 39 g/L) |
|---|---|---|---|---|---|---|
| 29/09/2021 | South Basin | N/A | 8.1 | 83 | 1.029 | 39,000 ppm (3.9%; 39 g/L) |
| 30/09/2021 | South Basin | N/A | 8.07 | 91 | 1.028 | 39,000 ppm (3.9%; 39 g/L) |
| 05/10/2021 | South Basin | N/A | 8.05 | 101 | 1.029 | 39,000 ppm (3.9%; 39 g/L) |
| 07/10/2021 | South Basin | 7.97 | 8.01 | 106 | 1.029 | 38,000 ppm (3.8%; 38 g/L) |
| 14/10/2021 | South Basin | 8.06 | 8.08 | 110 | 1.029 | 38,000 ppm (3.8%; 38 g/L) |
| 25/10/2021 | South Basin | N/A | 8.09 | 107 | 1.029 | 38,000 ppm (3.8%; 38 g/L) |
| 28/10/2021 | South Basin | 8.11 | 8.12 | 103 | 1.029 | 39,000 ppm (3.9%; 39 g/L) |
| 09/11/2021 | South Basin | 8.08 | 8.1 | 102 | 1.028 | 39,000 ppm (3.9%; 39 g/L) |
| 18/11/2021 | South Basin | 8.1 | 8.11 | 102 | 1.029 | 39,000 ppm (3.9%; 39 g/L) |
| 20/11/2021 | South Basin | N/A | 8.09 | 103 | 1.029 | 38,000 ppm (3.8%; 38 g/L) |
| 23/11/2021 | South Basin | N/A | 8.08 | 102 | 1.029 | 38,000 ppm (3.8%; 38 g/L) |
| 30/11/2021 | South Basin | 8.05 | 8.05 | 102 | 1.029 | 38,000 ppm (3.8%; 38 g/L) |
| 02/12/2021 | South Basin | N/A | 8.02 | 104 | 1.029 | 38,000 ppm (3.8%; 38 g/L) |
| 06/12/2021 | South Basin | N/A | 7.98 | 105 | 1.029 | 39,000 ppm (3.9%; 39 g/l) |
| 08/12/2021 | South Basin | N/A | 8.01 | 105 | 1.029 | 39,000 ppm (3.9%; 39 g/L) |
| 13/12/2021 | South Basin | N/A | 8.04 | 103 | 1.028 | 39,000 ppm (3.9%; 39 g/L) |
| 15/12/2021 | South Basin | 8.1 | 8.12 | 103 | 1.028 | 39,000 ppm (3.9%; 39 g/L) |
| 20/12/2021 | South Basin | N/A | 7.97 | 105 | 1.028 | 39,000 ppm (3.9%; 39 g/L) |
| 22/12/2021 | South Basin | N/A | 7.94 | 105 | 1.028 | 39,000 ppm (3.9%; 39 g/L) |
| 27/12/2021 | South Basin | N/A | 8.05 | 107 | 1.028 | 39,000 ppm (3.9%; 39 g/L) |
| 10/01/2022 | South Basin | 8.1 | 8.11 | 105 | 1.027 | 39,000 ppm (3.9%; 39 g/L) |
| 12/01/2022 | South Basin | 8.07 | 8.09 | 106 | 1.028 | 39,000 ppm (3.9%; 39 g/L) |
| 19/01/2022 | South Basin | 8.08 | 8.09 | 108 | 1.028 | 39,000 ppm (3.9%; 39 g/L) |
| 27/01/2022 | South Basin | 8.20 | 8.20 | 97 | 1.028 | 39,000 ppm (3.9%; 39 g/L) |

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
