# Peer review of "Open-Loop Exhaust-Gas Cleaning System: Analysis of Effects Produced on Barcelona Port Water pH"

_water, doi:10.3390/w14071146_

Round 1

Reviewer 1 Report

The work deals with very important aspects of exhaust gas cleaning in heavy industry. The summary does not explicitly mention the work's innovativeness. The introduction is done correctly. It contains a large number of citations. The work contains a good editorial content. This applies to all chapters. In my opinion, the quality of the drawings should be improved, the charts do not look too professional. The summary should be mentioned in points. It is good to provide numerical values. I would suggest a more extensive literature review. Attachments method is correct.

Author Response

Dear Sir,

We wish to submit a revised version of our paper entitled “OPEN-LOOP EXHAUST GAS CLEANING SYSTEM. ANALYSIS OF EFFECTS PRODUCED ON THE BARCELONA PORT WATER pH” for consideration by MDPI journals.

We inform to the editors and referees that, changes made to our manuscript are as follows:

  • The summary mentions the work's innovativeness and includes what is done in the paper and what is achieved.
  • The quality of the drawings has been improved. In the case of drawings from section 6 “Discussion” these have been deleted due to these drawings were intended to transmit a concept where all the buffering components were neutralized by an acid, but not were intended to give specific data about it.
  • The introduction has been improved by updating the literature review.
  • Spaces between pages have been revised.
  • There is a specific explanation regarding the variation in the parameters in Figures 5 to 9.
  • The novelty of the paper has been mentioned related to the literature review.
  • A description regarding the kind of ship and engine details has been included in section 4.
  • A more detailed explanation regarding the proposed measures to improve the exhaust gas cleaning system operation, has been included in section 7.

Thank you for your consideration of this manuscript.

Sincerely,

Reviewer 2 Report

The paper is interesting, while it is required to improve some parts as follows:

  1. the abstract must be improved, including what is done in the paper and what is achieved.
  2. the introduction required to be improved by updating the literature review including what is done in the previous years, especially in the last two years (there are a lot)
  3. Avoid big spaces between pages.
  4. The authors did not describe why there is variation in the parameters in Fig 5-9
  5. The figures and graphs can be improved, I think there is no need for colours as well as presenting the units.
  6. The novelty of the paper must be clearly mentioned related to the literature review.
  7. What are the characteristics of the ship and engine?
  8. The authors can describe in more detail what will be done in the next research and how they can improve the problem in water port.
  9. I think it will be interesting if the authors can add some parts related to the emissions in the air.

Author Response

Dear Sir,

We inform to the editors and referees that, changes made to our manuscript are as follows:

  • The summary mentions the work's innovativeness and includes what is done in the paper and what is achieved.
  • The quality of the drawings has been improved. In the case of drawings from section 6 “Discussion” these have been deleted due to these drawings were intended to transmit a concept where all the buffering components were neutralized by an acid, but not were intended to give specific data about it.
  • The introduction has been improved by updating the literature review.
  • Spaces between pages have been revised.
  • There is a specific explanation regarding the variation in the parameters in Figures 5 to 9.
  • The novelty of the paper has been mentioned related to the literature review.
  • A description regarding the kind of ship and engine details has been included in section 4.
  • A more detailed explanation regarding the proposed measures to improve the exhaust gas cleaning system operation, has been included in section 7.

Thank you for your consideration of this manuscript.

Sincerely,
